# Association between Serum Indoxyl Sulfate Levels and Endothelial Function in Non-Dialysis Chronic Kidney Disease

**DOI:** 10.3390/toxins11100589

**Published:** 2019-10-11

**Authors:** Chih-Hsien Wang, Yu-Hsien Lai, Chiu-Huang Kuo, Yu-Li Lin, Jen-Pi Tsai, Bang-Gee Hsu

**Affiliations:** 1Division of Nephrology, Hualien Tzu Chi Hospital, Buddhist Tzu Chi Medical Foundation, Hualien 97010, Taiwan; wangch33@gmail.com (C.-H.W.); hsienhsien@gmail.com (Y.-H.L.); hermit.kuo@gmail.com (C.-H.K.); nomo8931126@gmail.com (Y.-L.L.); 2School of Medicine, Tzu Chi University, Hualien 97004, Taiwan; 3Division of Nephrology, Department of Internal Medicine, Dalin Tzu Chi Hospital, Buddhist Tzu Chi Medical Foundation, Chiayi 62247, Taiwan

**Keywords:** chronic kidney disease, digital thermal monitoring test, endothelial function, indoxyl sulfate, vascular reactivity index

## Abstract

Indoxyl sulfate (IS), a product metabolized from tryptophan, is negatively correlated with renal function and cardiovascular diseases in patients with chronic kidney disease (CKD). We investigated the association between serum IS levels and endothelial function in patients with CKD. Fasting blood samples were obtained from 110 patients with stages 3–5 CKD. The endothelial function, represented by vascular reactivity index (VRI), was measured non-invasively using digital thermal monitoring. Serum IS levels were determined using liquid chromatography–mass spectrometry. Twenty-one (19.1%), 36 (32.7%), and 53 (48.2%) patients had poor (VRI < 1.0), intermediate (1.0 ≤ VRI < 2.0), and good (VRI ≥ 2.0) vascular reactivity. By univariate linear regression analysis, a higher prevalence of smoking, advanced age, higher systolic, and diastolic blood pressure (DBP), elevated levels of serum phosphorus, blood urea nitrogen, creatinine, and IS were negatively correlated with VRI values, but estimated glomerular filtration rate negatively associated with VRI values. After being adjusted by using multivariate stepwise linear regression analysis, DBP and IS levels were significantly negatively associated with VRI values in CKD patients. We concluded that IS level associated inversely with VRI values and had a modulating role in endothelial function in patients with stages 3–5 CKD.

## 1. Introduction

For a majority of patients with chronic kidney disease (CKD), cardiovascular disease (CVD) is the main cause of mortality not fully related to traditional risk factors such as age, hypertension (HTN), and diabetes mellitus (DM), but also non-traditional factors including inflammation, abnormal bone and mineral metabolism, and endothelial damage or dysfunction [1,2]. Endothelial dysfunction can be observed in all patients with CKD from pre-dialysis to maintenance hemodialysis (HD) and peritoneal dialysis (PD) [3,4,5,6]. Increasing evidence indicates that endothelial dysfunction can be measured non-invasively and is linked to adverse outcomes in patients with CKD [7,8]. Additionally, studies have shown that dyslipidemia, HTN, hyperparathyroidism, inflammation, oxidative stress (OS), and uremic toxins retention can modulate the process of endothelial dysfunction [4,9,10]. Recently, Naghavi et al. conducted a study in a registry of 6084 patients comparing different methods of measuring functional endothelial dysfunction, including digital thermal monitoring (DTM), flow-mediated dilatation, peripheral arterial tonometry, and photoplethysmography [11]. Digital thermal monitoring was shown to provide equivalent ability of measurements compared to other methods and was easier in terms of use and applicability to medical staff or patients [11].

Indoxyl sulfate (IS), initially only known as a gut-derived protein-bound uremic toxin, accumulates as renal function decreases and accompanied by a negative correlation with the level of kidney function [12]. Besides being a biomarker associated with decreased renal function, IS contributes to CKD progression through mechanisms including the induction of renal tubular damage or tubulointerstitial fibrosis by activating free radical production, upregulating nuclear factor (NF)-κB and plasminogen activator inhibitor type 1, and enhancing the expression of transforming growth factor beta 1, tissue inhibitor of metalloproteinase, and pro-alpha 1 collagen [13,14]. In addition, IS was reported to be related to aortic calcification, vascular stiffness, along with an increased risk of overall and cardiovascular (CV) mortality in patients with CKD through mechanisms of increasing OS in endothelial cells, shedding of endothelial microparticles, impairing endothelial cell repair, and inducing vascular smooth muscle cell proliferation [14]. Given that IS might have a role in endothelial dysfunction, which can cause CVD in patients with CKD, and we tried to apply non-invasive method to measure endothelial function, we conducted this study to identify the risk factors for endothelial dysfunction by using DTM in patients with stages 3–5 CKD.

## 2. Results

The clinical characteristics and medications used for patients with CKD are presented in Table 1. Among 110 patients with CKD, 21 (19.1%), 36 (32.7%), and 53 (48.2%) were individually diagnosed with poor, intermediate, and good VRI, respectively. Patients who were older had poorer vascular reactivity (*p* = 0.020). Systolic blood pressure (SBP, *p* = 0.003), diastolic blood pressure (DBP, *p* = 0.004) and serum blood urea nitrogen (BUN, *p* = 0.001), creatinine (*p* = 0.001), phosphorus (*p* = 0.016), and IS (*p* < 0.001) levels increased significantly, whereas estimated glomerular filtration rate (eGFR, *p* < 0.001) decreased significantly as VRI decreased among patients with CKD. With respect to renal function defined by CKD stage, there were more patients at more advanced CKD stages with lower VRI values (*p* < 0.001). The proportion of patients with a smoking habit also increased as VRI decreased (*p* = 0.022). Among the patients with CKD, 54 had DM (49.1%) and 91 had HTN (82.7%). Prescribed medications for the patients included angiotensin receptor blocker (ARB, n = 57; 51.8%), β-blocker (n = 35; 31.8%), α-blocker (n = 20; 18.2%), calcium channel blocker (CCB, n = 46; 41.8%), statin (n = 55; 50.0%), and fibrate (n = 9; 8.2%). There were no significant differences with respect to sex, the presence of DM and HTN, or the use of medications among the three groups of patients.

Advanced age (*r* = −0.264, *p* = 0.005), SBP (*r* = −0.243, *p* = 0.011), DBP (*r* = −0.233, *p* = 0.014), smoking habit (*r* = −0.207, *p* = 0.030), serum phosphorus level (*r* = −0.228, *p* = 0.017), logarithmically transformed BUN (log-BUN, *r* = −0.337, *p* < 0.001), log-creatinine (*r* = −0.386, *p* < 0.001), and log-IS level (*r* = −0.700, *p* < 0.001) were negatively correlated, whereas eGFR (*r* = 0.350, *p* < 0.001) was positively correlated with VRI values in patients with CKD by univariate linear regression analysis (Table 2). In addition, adjustments for variables significantly associated with VRI values in univariate linear regression analysis revealed that DBP (standardized β = −0.142, adjusted R^2^ change = 0.016; *p* = 0.040) and serum log-IS level (standardized β = −0.681, adjusted R^2^ change = 0.485; *p* < 0.001) were significantly and independently negatively associated with VRI values in patients with CKD by multivariate stepwise linear regression analysis (Table 2). The two-dimensional scattered plots of VRI values with DBP and serum log-IS level among these patients with stages 3–5 CKD are presented in Figure 1a,b.

## 3. Discussion

This study showed that advanced age, smoking and serum phosphorus, BUN, creatinine, SBP, DBP, and IS levels were negatively associated with VRI values assessed by DTM among patients with CKD. Furthermore, after adjusting for confounders, the levels of VRI were significantly and independently associated with DBP and serum log-IS level in patients with CKD.

Endothelial damage, along with traditional risk factors such as age, HTN, and DM, is considered a progressive factor of CVD in patients with CKD [1,2]. In a long-term observational study, atherosclerosis, which presents with endothelial dysfunction as an early marker, deteriorated in the uremic state; thus, atherosclerosis is considered as the main cause of CVD in patients with CKD [15]. The endothelium not only serves as an inner layer lining the vascular conduit, which is involved in cellular and nutrient trafficking, but also mediates vascular activity that can be functionally assessed non-invasively or by examining biomarkers over the past two decades [11,14]. In patients with CKD, the circulating endothelin, thrombomodulin, and von Willebrand factor, which are indicative of endothelial function, are abnormally elevated [16,17]. Additionally, endothelial damage was reported to be an injury response, with mechanisms including endothelium-dependent vasodilation or decreased bioavailability of nitric oxide (NO) [6]. Mallamaci el al. reported that one of the endogenous inhibitors of nitric oxide synthase (NOS), asymmetrical dimethylarginine, could be considered a biomarker of endothelial dysfunction and may predict all-cause and CV mortality as well as serum brain natriuretic peptide or C-reactive protein levels in patients with HD [18]. Studies have shown that endothelial function, assessed respectively with the reactive hyperemia peripheral arterial tonometry index or brachial artery ultrasound, correlates with the presence of coronary artery disease and portends the future development of CV events in patients with CKD and peripheral arterial disease [7,8]. The endothelium is a target of injury even in renal transplantation patients. We recently found that endothelial dysfunction measured by VRI, was correlated with serum adipocyte fatty acid binding protein, which is a potential biomarker of CVD [19]. Here in our research, we similarly assessed endothelial function by using a non-invasive method, the DTM, and found that as the renal function worsened, the values of VRI decreased. Altogether, these studies indicate that as the renal function of patients with CKD worsens, so too does their endothelial function.

IS, accumulated as the decline of renal function, also contributes to renal damage [13,14]. In addition to playing a role in the progression of renal function, growing evidence has shown that IS can cause CVD in patients with CKD or end-stage renal disease [20,21,22]. Barreto et al. reported that serum IS levels are not only significantly positively associated with image-graded aortic calcification and pulse wave velocity but also predicted the overall and CV mortality in patients with CKD [20]. Studies have demonstrated that serum IS is a valuable marker to predict CV and dialysis events in patients with advanced CKD and heart failure events in patients with ESRD [21,22]. In-vitro studies with human umbilical vein endothelial cells revealed that IS inhibited endothelial proliferation and wound repair and induced ROS through the induction of nicotinamide adenine dinucleotide phosphate oxidase to inhibit NO production and viability [23,24,25]. An oral adsorbent, well known to reduce serum IS levels, was reported to decrease carotid artery intima media thickness and improve pulse wave velocity and flow-mediated vasodilatation with a decrease in the IS levels and oxidized/reduced glutathione ratio of patients with CKD [25,26]. In one systemic review, IS was found to induce endothelial dysfunction and aortic stiffness by enhancing ROS, impairing endothelial cell repair, and inducing vascular smooth muscle cell proliferation, resulting in an increased risk of overall and CV mortality in patients with CKD [27]. Moreover, Ito et al. found that aryl hydrocarbon receptor, which was known as a transcription factor modulating toxic effects of uremic toxins, could dysregulate IS-enhanced leukocyte adhesion as well as E-selectin expression through modulating activator protein-1 transcriptional activities in mice vascular endothelial cells [28]. Due to these findings, IS is regarded as the main risk factor for endothelial dysfunction through mechanisms such as activation of oxidative stress aryl hydrocarbon receptor or impaired vasodilatation in patients with CKD. We similarly found that the levels of IS correlated negatively with the endothelial function, presented as VRI in patients with CKD in this study.

In a community study in which endothelium function was measured using several non-invasive methods, Schnabel et al. reported that advanced age as well as high blood pressure were consistently thought to be risk factors [29]. Age-associated endothelial dysfunction, most commonly associated with impaired endothelium-dependent dilation, was shown in vivo to be mediated by reduced NO bioavailability and increased production of ROS [30]. Naghavi et al. and our previous study showed that along with elevated DBP, advanced age was an independent predictor of VRI, measured by digital thermal monitoring in the general population and in kidney transplantation patients [11,20]. In the present study, we similarly found that both advanced age and high DBP were inversely correlated with VRI values in patients with CKD.

Hyperphosphatemia not only progressively elevates as renal function decreases but also stimulates phenotypic changes in vascular smooth cells to differentiate into osteoblast-like cells with abnormal mineral deposition on vascular walls [1,2]. In addition, after the incubation of endothelial cells with high concentrations of phosphate, the integrity of the cells is impaired, resulting in apoptosis through the activation of ROS [31]. After being stimulated by high phosphate, human umbilical vein endothelial cells exhibited lower endothelial NOS and NO production, along with an increased number of apoptotic and necrotic cells [32]. Taken together, hyperphosphatemia could both induce endothelial dysfunction and stimulate vascular calcification [33,34]. Smoking is also a cardiovascular risk factor that induces ROS production, inflammation, and endothelial dysfunction [34]. We similarly found that endothelial dysfunction correlated with serum phosphate and smoking in patients with CKD.

This study has some limitations. First, this was a cross-sectional study. Second, the study enrolled a limited number of patients with CKD at a single hospital. Third, this study only included patients with stages 3–5 CKD and patients with CKD stage 1 or 2 and dialysis patients were excluded. Fourth, this study could not extrapolate to other races because we enrolled only Asian patients. Fifth, albuminuria is associated with endothelial dysfunction and is also a cardiovascular risk factor in patients with and without diabetes [35]. However, our study did not measure albuminuria. Finally, because statistical significance in observational studies could simply imply association but could not confirm biological significance. Therefore, the causal relationship between serum IS levels and endothelial function should be investigated with more patients using a longitudinal method.

## 4. Conclusions

Along with DBP, endothelial dysfunction, which is represented by low VRI values, is positively correlated with the serum levels of IS in patients with CKD stages 3–5.

## 5. Materials and Methods

### 5.1. Participants

A total of 110 patients with CKD were enrolled from October 2017 to February 2018 at a medical center in Hualien, Taiwan, After reviewing their medical records, HTN or DM were respectively defined as SBP ≥ 140 mmHg and/or DBP ≥ 90 mmHg, the prescription of any anti-HTN medications or fasting plasma glucose level ≥ 126 mg/dL, or the administration of oral hypoglycemic medications or insulin. The body weights and heights of the participants were measured in light clothing and without shoes to the nearest 0.5 kilograms and 0.5 cm, respectively. Body mass index (BMI) was expressed as weight (kg)/height(m)^2^. We excluded patients who had an acute infection, acute myocardial infarction, heart failure, chronic obstructive pulmonary disease, and malignancy at the time of blood sampling, or if they refused to provide informed consent. The Protection of the Human Subjects Institutional Review Board of Tzu Chi University and Hospital approved this study (IRB106-108-A) and were carried out following the rules of the Declaration of Helsinki. All patients were required to provide informed consent before being investigated in this study.

### 5.2. Biochemical Analysis and CKD Stage

Fasting blood samples (approximately 5 mL) of patients were immediately centrifuged and examined by an auto-analyzer (Siemens Advia 1800, Siemens Healthcare GmbH, Henkestr, Germany) for the measurements of serum BUN, creatinine, albumin, globulin, fasting glucose, total cholesterol, triglycerides, low-density lipoprotein cholesterol, total calcium, and phosphorus levels. The Chronic Kidney Disease Epidemiology Collaboration (CKD-EPI) equation was applied to calculate eGFR using the mean value of measurements at least 3 months apart. The participants were divided into different CKD stages according to the Kidney Disease Outcomes Quality Initiative criteria [36]. Patients were considered to have CKD stage 3, 4, or 5 if eGFR = 59–30 mL/min per 1.73 m^2^, 29–15 mL/min per 1.73 m^2^, or <15 mL/min per 1.73 m^2^, individually.

### 5.3. Determination of Serum IS Levels

This study used a Waters e2695 HPLC system consisting of a single quadrupole mass spectrometer (ACQUITY QDa, Waters Corporation, Milford, MA, USA). The analytical column was Phenomenex Luna^®^ C18(2) (Phenomenex, Torrance, CA, USA) (5 µ, 250 × 4.60 mm, 100 Å); the temperature, flow, and injection were respectively set at 40 °C, 0.8 mL/min, and 30 µL. The mobile phase consisted of a binary gradient applied as follows: the initial composition (95% (A) water + 0.1% formic acid/5% (B) methanol + 0.1% formic acid) was kept constant for 1 min; solvent B was then increased linearly up to 70% over 12 min and kept constant for 2 min. For column re-equilibration, solvent B was reduced to 50% over 1 min and kept constant for 2 min.

Liquid chromatography–mass spectrometry analysis of the serum IS level was performed using a modified method [37,38]. Pretreated samples were synchronously assessed in negative (i.e., IS)-ion mode electrospray ionization. The instrument settings were as follows: desolvation temperature, 600 °C; capillary voltage, 0.8 kV; and sample cone, 15.0 V. The mass spectrometer was operated in full scan 50–450 m/z for positive-ion mode and 100–350 m/z for negative-ion mode. The single-ion recording mode was used to monitor the individual masses of each compound (IS: 211.9 m/z). Endogenous compounds were quantified by measuring the peak areas and comparing the values with a calibration curve obtained from standard solutions. Empower^®^ 3.0 software (New York, NY, USA) was used for data acquisition and processing. The retention time for IS was approximately 13.86 min. According to Shu et al. [37], this method of measuring IS levels had good performance with the relative standard deviation of intra- and inter-day precision within ± 15%.

### 5.4. Endothelial Function Measurements

After overnight fasting and resting supinely at an ambient temperature of 22 °C–24 °C for 30 minutes, BP cuffs were placed on the subject’s right upper arm and skin temperature sensors were affixed to both of the subject’s index fingers. An FDA-approved device (VENDYS-II, Endothelix Inc., Houston, TX, USA) with DTM for the measurements of the endothelial function was applied to the patients [20]. The measurements of DTM performed with both hands during 5-minute stabilization, 5-minutes cuff inflation to 50 mmHg > SBP, and 5-minute deflation. As soon as the cuff was deflated, blood flow rushed into the forearm and hand, causing a temperature rebound in the fingertip, which was directly proportional to the reactive hyperemia response. VRI was determined by taking the maximum difference between the observed temperature rebound curve and the zero-reactivity curve during the reactive hyperemia period using VENDYS software. Patients were defined as having poor, intermediate, or good VRI if VRI was 0.0 to < 1.0, 1.0 to < 2.0, or ≥ 2.0, respectively [11,20].

### 5.5. Statistical Analysis

Continuous variables were analyzed by the Kolmogorov–Smirnov test, and variables were expressed as means ± standard deviation or medians with interquartile ranges depending on whether they exhibited normal distribution. Differences among groups (poor, intermediate, and good VRI) were analyzed using Kruskal–Wallis analysis or one-way analysis of variance followed by the Fisher’s protected t test for parameters with and without normal distribution. Because fasting glucose, triglyceride, BUN, creatinine, globulin, total calcium, and IS levels were not shown to have normally distribution, these variables were logarithmically transformed before further analysis. Variables correlated with VRI values were evaluated by univariate linear regression analysis and variables with significant results in the univariate linear regression analysis, including age, smoking, SBP, DBP, Log-BUN, Log-creatinine, eGFR, phosphorus and Log-IS were further analyzed with multivariate stepwise linear regression analysis. SPSS for Windows was used for analysis (version 19.0; SPSS Inc., Chicago, IL, USA). A *p*-value of < 0.05 was considered statistically significant.

## Figures and Tables

**Figure 1 toxins-11-00589-f001:**
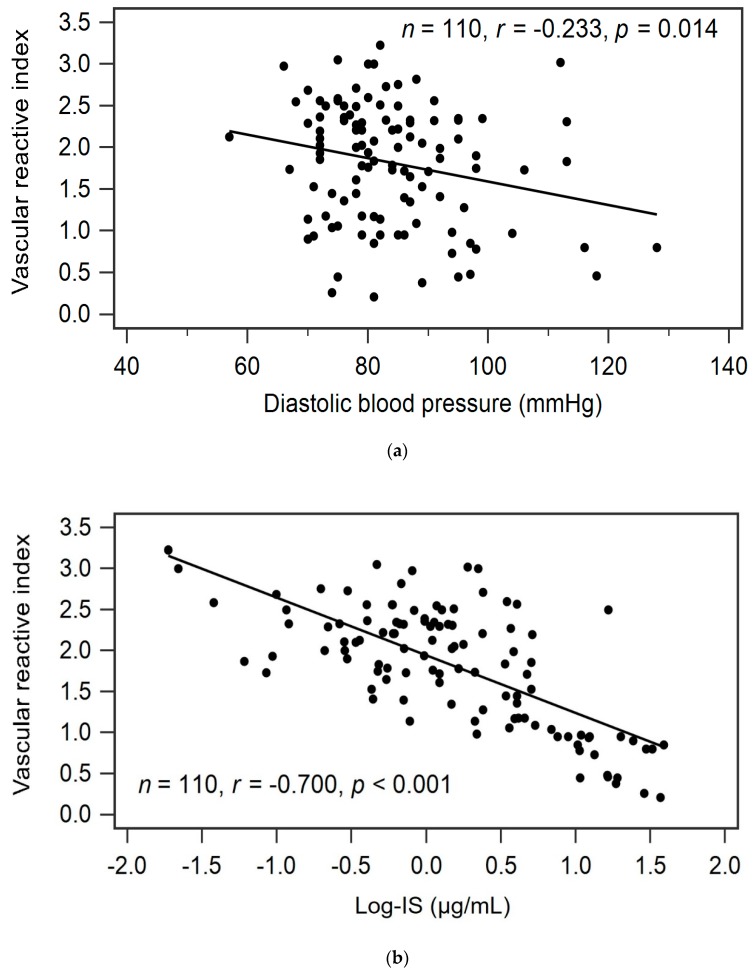
Relationships between vascular reactive index and (**a**) diastolic blood pressure (mmHg), or (**b**) log-IS level (μg/mL) among 110 stage 3–5 patients with chronic kidney disease.

**Table 1 toxins-11-00589-t001:** Clinical characteristics of 110 patients with chronic kidney disease according to vascular reactivity index measured with digital thermal monitoring.

Characteristics	All Patients (n = 110)	Good Vascular Reactivity (n = 53)	Intermediate Vascular Reactivity (n = 36)	Poor Vascular Reactivity (n = 21)	*p* Value
VRI	1.82 ± 0.72	2.44 ± 0.30	1.55 ± 0.29	0.72 ± 0.26	< 0.001 *
Age (years)	65.89 ± 7.94	64.09 ± 8.40	66.31 ± 6.99	69.71 ± 7.07	0.020 *
Female, n (%)	50 (45.5)	26 (49.1)	13 (36.1)	11 (52.4)	0.377
Height (cm)	159.33 ± 9.04	160.08 ± 9.23	158.71 ± 9.13	158.50 ± 8.64	0.705
Body weight (kg)	67.55 ± 14.12	67.62 ± 14.30	68.06 ± 13.75	66.51 ± 14.89	0.924
BMI (kg/m^2^)	26.51 ± 4.66	26.32 ± 4.97	26.94 ± 4.45	26.25 ± 4.34	0.796
Diabetes mellitus, n (%)	54 (49.1)	28 (52.8)	15 (41.7)	11 (52.4)	0.554
Hypertension, n (%)	91 (82.7)	42 (79.2)	29 (80.6)	20 (95.2)	0.238
Smoking, n (%)	13 (11.8)	3 (5.7)	4 (11.1)	6 (28.6)	0.022 *
SBP (mmHg)	145.65 ± 24.24	139.94 ± 20.39	145.28 ± 20.31	160.71 ± 32.78	0.003 *
DBP (mmHg)	83.86 ± 11.94	81.15 ± 10.17	83.61 ± 10.33	91.14 ± 15.69	0.004 *
BUN (mg/dL)	31.00 (23.00–45.75)	27.00 (22.50–38.00)	32.00 (24.25–49.75)	63.00 (31.00–75.00)	0.001 *
Creatinine (mg/dL)	1.75 (1.40–2.50)	1.70 (1.30–2.35)	1.75 (1.43–2.20)	3.70 (2.20–5.70)	0.001 *
eGFR (mL/min)	33.34 ± 15.76	37.93 ± 13.96	34.51 ± 15.19	19.74 ± 13.85	< 0.001 *
CKD stage 3, n (%)	63 (57.3)	37 (69.8)	21 (58.3)	5 (23.8)	< 0.001 *
CKD stage 4, n (%)	27 (24.5)	13 (24.5)	9 (25.0)	5 (23.8)	
CKD stage 5, n (%)	20 (18.2)	3 (5.7)	6 (16.7)	11 (52.4)	
Total cholesterol (mg/dL)	159.54 ± 47.24	158.68 ± 47.71	150.75 ± 42.33	176.76 ± 51.56	0.132
Triglyceride (mg/dL)	124.50 (93.50–172.00)	119.00 (94.50–168.50)	123.00 (89.25–160.25)	131.00 (99.50–213.50)	0.396
LDL-C (mg/dL)	90.38 ± 38.96	92.43 ± 42.01	82.06 ± 32.69	99.48 ± 39.88	0.232
Fasting glucose (mg/dL)	105.00 (98.00–141.00)	105.00 (99.50–140.50)	101.00 (97.00–139.00)	109.00 (96.50–160.50)	0.635
Albumin (mg/dL)	4.11 ± 0.39	4.16 ± 0.42	4.08 ± 0.29	4.05 ± 0.46	0.512
Globulin (mg/dL)	2.90 (2.80–3.20)	2.90 (2.80–3.15)	3.00 (2.80–3.16)	2.90 (2.70–3.25)	0.932
Total calcium (mg/dL)	8.88 (8.60–9.25)	8.88 (8.72–9.28)	8.90 (8.51–9.31)	8.80 (8.46–9.06)	0.231
Phosphorus (mg/dL)	3.89 ± 0.95	3.68 ± 0.64	3.91 ± 0.94	4.38 ± 1.39	0.016 *
Indoxyl sulfate (μg/mL)	1.33 (0.53–4.58)	0.70 (0.32–1.49)	1.56 (0.54–4.02)	16.32 (10.63–26.44)	< 0.001 *
ARB use, n (%)	57 (51.8)	28 (52.8)	18 (50.0)	11 (52.4)	0.965
β-blocker use, n (%)	35 (31.8)	13 (24.5)	11 (30.6)	11 (52.4)	0.067
α-blocker use, n (%)	20 (18.2)	7 (13.2)	7 (19.4)	6 (28.6)	0.295
CCB use, n (%)	46 (41.8)	20 (37.7)	13 (36.1)	13 (61.9)	0.115
Statin use, n (%)	55 (50.0)	29 (54.7)	15 (41.7)	11 (52.4)	0.468
Fibrate use, n (%)	9 (8.2)	5 (9.4)	3 (8.3)	1 (4.8)	0.803

Continuous variables are shown as means ± standard deviation and tested by one-way analysis of variance; variables not with normal distribution are shown as medians and interquartile range and tested by Kruskal–Wallis analysis; categorical variables are shown as number (%) and analyzed by the chi-square test. BMI, body mass index; VRI, vascular reactivity index; SBP, systolic blood pressure; DBP, diastolic blood pressure; LDL-C, low-density lipoprotein cholesterol; BUN, blood urea nitrogen; eGFR, estimated glomerular filtration rate; ARB, angiotensin receptor blocker; CCB, calcium channel blocker. * *p* < 0.05 was considered statistically significant.

**Table 2 toxins-11-00589-t002:** Correlation of vascular reactivity index levels and clinical variables by univariate or multivariate stepwise linear analyses in 110 patients with chronic kidney disease.

Variables	Vascular Reactivity Index				
	Univariate		Multivariate		
	*r*	*p* Value	Standardized Beta	Adjusted R^2^ Change	*p* Value
Female	0.085	0.379	−	−	−
Diabetes mellitus	−0019	0.847	−	−	−
Hypertension	−0.070	0.466	−	−	−
Smoking	−0.207	0.030 *	−	−	−
Age (years)	−0.264	0.005 *	−	−	−
Height (cm)	0.018	0.853	−	−	−
Body weight (kg)	−0.007	0.945	−	−	−
Body mass index (kg/m^2^)	−0.016	0.868	−	−	−
Systolic blood pressure	−0.243	0.011 *	−	−	−
Diastolic blood pressure	−0.233	0.014 *	−0.142	0.016	0.040 *
Total cholesterol (mg/dL)	0.017	0.857	−	−	−
Log-Triglyceride (mg/dL)	−0.106	0.271	−	−	−
LDL-C (mg/dL)	0.067	0.486	−	−	−
Log-Glucose (mg/dL)	0.002	0.983	−	−	−
Log-BUN (mg/dL)	−0.337	< 0.001 *	−	−	−
Log-Creatinine (mg/dL)	−0.386	< 0.001 *	−	−	−
eGFR (mL/min)	0.350	< 0.001 *	−	−	−
Albumin (mg/dL)	0.097	0.311	−	−	−
Log-Globulin (mg/dL)	0.054	0.579	−	−	−
Log-Calcium (mg/dL)	0.126	0.190	−	−	−
Phosphorus (mg/dL)	−0.228	0.017 *	−	−	−
Log-IS (μg/mL)	−0.700	< 0.001 *	−0.681	0.485	< 0.001 *

Values of triglyceride, fasting glucose, blood urea nitrogen, creatinine, globulin, calcium, and IS did not show normal distribution and were log-transformed for further analysis. Correlation between variables and VRI was analyzed by using univariate linear regression analysis or multivariate stepwise linear regression analysis (adjusted by smoking, age, systolic blood pressure, diastolic blood pressure, log-BUN, log-Creatinine, eGFR, phosphorous, and log-IS). BUN, Blood urea nitrogen; LDL-C, low-density lipoprotein cholesterol; eGFR, estimated glomerular filtration rate; IS, indoxyl sulfate. * *p* < 0.05 was considered statistically significant.

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
