# Peer review of "Association between Serum Indoxyl Sulfate Levels and Endothelial Function in Non-Dialysis Chronic Kidney Disease"

_toxins, 2019, doi:10.3390/toxins11100589_

Round 1

Reviewer 1 Report

Interesting article, of value to the readers of toxins.

Major comments:

In table 2, the multivariable regression is performed using stepwise linear regression. In the methods sections it is not described how this is performed. Did you perform stepwise backward elimination? Please describe this more in the methods section.

In table 2, why did you report beta values and not standardized beta values. With standardized betas it is easier to check the strengths of the associations.

In table 2, you do analyses for both creatinine and eGFR. These variables can be expected to be very collinear with each other. Do the results change if use only one of these for the stepwise linear regression analyes?

How is the distribution of race in your study. Previous studies have, for instance, found a positive association between IS and percentage black. If your study is predominantly performed in Asian patients, please add that it is unclear whether these results can be extrapolated to other races.

In general, statistical significance in observational studies suggests, but does not confirm, biologic significance. Whether the significant relation between VRI and IS is a causal or an associative relation remains to be determined. Please add this as a limitation.

Please state in the methods section whether you did or did not use an internal standard for the LC-MS analysis of IS.

What was the inter and intra precision (coefficient of variation) of the IS method?

Minor comments:

Line 26: This sentence is very long and does not read fluently. Please shorten.

Line 36: In this sentence, the same thing is said twice.

Line 38: This indicates that IS used to measure decrease in renal function, while this is not the case.

Line 42: This sentence stretches over 6 lines and does not read fluently. Please shorten.

In table 1: Do you only have active smokers, or also ex-smokers?

Figure 1: Unnecessary space between micro and g/mL in the x-axis.

Line 203: Please state whether eGFR is based on creatinine, cystatin C or both.

Author Response

Comments and Suggestions for Authors

Interesting article, of value to the readers of toxins.

Major comments:

In table 2, the multivariable regression is performed using stepwise linear regression. In the methods sections it is not described how this is performed. Did you perform stepwise backward elimination? Please describe this more in the methods section.

Ans: Thanks for your recommendation. The sentence was revised as “Variables correlated with VRI values were evaluated by simple linear regression analysis and variables with significant results in the simple linear regression analysis, including smoking, age, SBP, DBP, Log-BUN, Log-creatinine, eGFR, phosphorus, and Log-IS were further analyzed with a multivariate stepwise linear regression analysis.”

In table 2, why did you report beta values and not standardized beta values. With standardized betas it is easier to check the strengths of the associations.

 Ans: Thanks for your recommendation. We did report the results using standardized beta instead of beta, but improperly described. We wound revised the description as standardized beta in Table 2 and the Results section.

In table 2, you do analyses for both creatinine and eGFR. These variables can be expected to be very collinear with each other. Do the results change if use only one of these for the stepwise linear regression analyes?

 Ans: Thanks for your recommendations. If we use multivariable stepwise linear regression analysis without using Log-Cre or eGFR individually, DBP and Log-IS remained significantly negatively associated with VRI.

How is the distribution of race in your study. Previous studies have, for instance, found a positive association between IS and percentage black. If your study is predominantly performed in Asian patients, please add that it is unclear whether these results can be extrapolated to other races.

 Ans: This study was conducted in a Taiwan medical center. So, this study could not extrapolate to other races. We would describe this as study limitation in the Discussion section. Thanks for your recommendation.

In general, statistical significance in observational studies suggests, but does not confirm, biologic significance. Whether the significant relation between VRI and IS is a causal or an associative relation remains to be determined. Please add this as a limitation.

 Ans:  Thanks for your wonderful recommendation. We will revise the description in the Discussion section as “Finally, because statistical significance in observational studies could simply imply association but could not confirm biological significance. Therefore, the causal relationship between serum IS levels and endothelial function should be investigated with more patients using a longitudinal method.”

Please state in the methods section whether you did or did not use an internal standard for the LC-MS analysis of IS. What was the inter and intra precision (coefficient of variation) of the IS method?

Ans: We did use an internal standard for the LC-MS analysis of IS and revised the description in the Materials and Methods section, Determination of serum IS levels, 2nd paragraph, Line 245, as “Endogenous compounds were quantified by measuring the peak areas and comparing the values with a calibration curve obtained from standard solutions. The retention time for IS was approximately 13.86 min. According to Shu et al. this method of measuring IS had good performance with the relative standard deviation of intra- and inter-day precision within ±15%” Thanks for your recommendation.

Minor comments:

Line 26: This sentence is very long and does not read fluently. Please shorten.

Ans: This sentence was revised as “By simple linear regression analysis, a higher prevalence of smoking, advanced age, higher systolic and diastolic blood pressure (DBP), elevated levels of serum phosphorus, blood urea nitrogen, creatinine, and IS were negatively correlated with VRI values, but estimated glomerular filtration rate negatively associated with VRI values. After adjusting, DBP and IS levels were significantly negatively associated with VRI values in CKD patients.” Thanks for your recommendations.

Line 36: In this sentence, the same thing is said twice.

Ans: We revised the description as “Key contribution: Serum IS levels are associated with poorer endothelial function in patients with stage 3-5 CKD.”

Line 38: This indicates that IS used to measure decrease in renal function, while this is not the case.

Ans: Thanks for your recommendations. We revised the description as “Besides being a biomarker associated with decreased renal function, IS contributes to CKD progression through mechanisms…”

Line 42: This sentence stretches over 6 lines and does not read fluently. Please shorten.

Ans: Thanks for your recommendations. This sentence was revised as “In addition, IS was reported to be related to aortic calcification, vascular stiffness, along with an increased risk of overall and cardiovascular (CV) mortality in patients with CKD through mechanisms of increasing OS in endothelial cells, shedding of endothelial microparticles, impairing endothelial cell repair, and inducing vascular smooth muscle cell proliferation.”

In table 1: Do you only have active smokers, or also ex-smokers?

Ans: Thanks for your recommendations. We only enrolled patients who are active smokers.

Figure 1: Unnecessary space between micro and g/mL in the x-axis.

Ans: Thanks for your suggestion. We revised the figure according to your recommendation.

Line 203: Please state whether eGFR is based on creatinine, cystatin C or both.

Ans: We calculated eGFR by CKD-EPI (Chronic Kidney Disease Epidemiology Collaboration) equation based on serum creatinine which is mentioned in Materials and Methods, Biochemical analysis and CKD stage section. Thanks for your recommendation.

Reviewer 2 Report

The authors have investigated possible association between serum indoxyl sulfate levels and endothelial function in non-dialysis chronic kidney disease. It was concluded that the indoxyl sulfate level was negatively associated with vascular reactivity index values and plays a role in modulating endothelial function in patients with stages 3–5 CKD.

The presented study was well designed and the obtained results support the author’s hypothesis.  Enrollment of participants was performed by following clear inclusion and exclusion criteria. Bioethical standards were followed. The methodology used was appropriately chosen, and obtained results were adequately analyzed. Limitations of the study were clearly disclosed.

Author Response

Comments and Suggestions for Authors

The authors have investigated possible association between serum indoxyl sulfate levels and endothelial function in non-dialysis chronic kidney disease. It was concluded that the indoxyl sulfate level was negatively associated with vascular reactivity index values and plays a role in modulating endothelial function in patients with stages 3–5 CKD.

The presented study was well designed and the obtained results support the author’s hypothesis.  Enrollment of participants was performed by following clear inclusion and exclusion criteria. Bioethical standards were followed. The methodology used was appropriately chosen, and obtained results were adequately analyzed. Limitations of the study were clearly disclosed.

Ans: Thanks for your comments. Hopefully, this manuscript will be acceptable to you and other reviewers. Your favorable considerations will be highly appreciated.

Reviewer 3 Report

-In this study, the link between indoxyl sulfate levels and endothelial function in patients with chronic kidney disease was investigated. The hypothesis and objective of the study is very interesting and is an important question in the Nephrology field; however, given the type of this study, there are major concerns about the statistical analysis which needs to be addressed by the authors. 

-Overall punctuation and grammar errors. 

-Line 55 Results: Please re-phrase the sentence as age cannot change with the change in VRI. 

-Line 67 Table 1: Please consider changing how the statistics has been done for this table and how the table is reported. It is not clear if participants were randomly chosen to be in each group or they were selected based on their clinical outcomes...There is no balance between groups and as it is mentioned, there is a significant differences between groups as far as age. However, authors should either stratify based on their groups or match their groups so they would be similar with regards to their demographics such as age, sex, medications, etc. 

-Line 67 Table 1: Please separate demographics from clinical characteristics.

-Line 76 Results: Please either report on raw or adjusted results. 

-Line 189 Methods: There is a major issue with how the patients have been selected. For instance, a DM patients who is taking insulin may have significantly different clinical outcome compared to a pre-diabetic patient and compared with early stages of HTN. Please consider changing how you included the participants or stratify based on your selected patients. 

-Line 205 Methods: If patients are divided based on the stage of CKD, then please report the results based on that as it is not clear in the results how the stage of CKD played any roles in the reported results. 

Author Response

Comments and Suggestions for Authors

-In this study, the link between indoxyl sulfate levels and endothelial function in patients with chronic kidney disease was investigated. The hypothesis and objective of the study is very interesting and is an important question in the Nephrology field; however, given the type of this study, there are major concerns about the statistical analysis which needs to be addressed by the authors. 

-Overall punctuation and grammar errors.

Ans: We have revised the whole manuscript from an English revision service and would upload the certificate of English editing. Thanks for your recommendation.

 -Line 55 Results: Please re-phrase the sentence as age cannot change with the change in VRI.

Ans: This sentence was revised as “Patients who were older wound have poorer vascular reactivity (p = 0.020). Systolic blood pressure (SBP, p = 0.003), diastolic blood pressure (DBP, p = 0.004) and serum blood urea nitrogen (BUN, p = 0.001), creatinine (p = 0.001), phosphorus (p = 0.016), and IS (p < 0.001) levels increased significantly, whereas estimated glomerular filtration rate (eGFR, p < 0.001) decreased significantly as VRI decreased among patients with CKD.” Thanks for your comments.

 -Line 67 Table 1: Please consider changing how the statistics has been done for this table and how the table is reported. It is not clear if participants were randomly chosen to be in each group or they were selected based on their clinical outcomes...There is no balance between groups and as it is mentioned, there is a significant differences between groups as far as age. However, authors should either stratify based on their groups or match their groups so they would be similar with regards to their demographics such as age, sex, medications, etc. 

Ans: In this study, we enrolled patients from only one medical center in Hualien with limited duration of time (from Oct. 2017 to Feb. 2018). After excluding those with active acute infection, AMI, heart failure, COPD, or malignancy, there were totally 110 CKD patients provide informed consents and agreed to participate in this study. Because limited number of patients and limited duration of time enrolling patients, we divided patients into three categories according to their VRI values instead of by a balance method and aimed to analyze the possible factors, including demographic or biochemical factors, that could be associated with lower VRI by using multivariable stepwise linear regression analysis. Thanks for your comments.

-Line 67 Table 1: Please separate demographics from clinical characteristics.

Ans: We re-arranged Table 1 according to your recommendation. Thanks for your suggestion.

-Line 76 Results: Please either report on raw or adjusted results. 

Ans: Results of both univariate and multivariate linear regression analyses were described in this Results section. Thanks for your comments.

-Line 189 Methods: There is a major issue with how the patients have been selected. For instance, a DM patients who is taking insulin may have significantly different clinical outcome compared to a pre-diabetic patient and compared with early stages of HTN. Please consider changing how you included the participants or stratify based on your selected patients. 

Ans: In this study, we enrolled a totally of 110 CKD patients if they provide informed consents and agreed to participate in this study after excluding those with active acute infection, AMI, heart failure, COPD, or malignancy, there were. As we described in Discussion section, DM and HTN surely could cause endothelial damage. However, our patients divided based on VRI did not have significantly difference of having DM or HTN or even the usage of medications, which included ARB, beta-blocker, alpha-blocker, CCB, statin or fibrate. Moreover, we would consider stratify in future study based on your recommendations if the number of patients could be more. Thanks for your recommendations.

-Line 205 Methods: If patients are divided based on the stage of CKD, then please report the results based on that as it is not clear in the results how the stage of CKD played any roles in the reported results. 

Ans: Thanks for your recommendations. Because we aimed to analyze the risk factors of developing endothelial dysfunction, so our patients were divided based on values of VRI. From Table 1, we reported that as renal function decreased (increased serum levels of BUN, creatinine and decreased eGFR), values of VRI decreased. To make Table 1 clearer showing renal function according to CKD stage, the percentage of patients (categorized into three groups based on VRI) in different stages of CKD were reported. In Table 2, we only reported the association of decreasing renal function and poorer VRI by using continuous variables as BUN, Creatinine and eGFR instead of categorical variable (CKD stage). Although we did not show the results of the association between the CKD stage and VRI, the CKD stage was also associated with VRI after analysis (r = -0.364, p = 0.0001, data not shown)

Reviewer 4 Report

The manuscript describes a clinical study in which the association between serum indoxyl sulfate levels and endothelial function in non-dialysis CKD patients was investigated. Already multiple studies (in vitro and in vivo studies) have shown that indoxyl sulfate induces deleterious effects on the endothelial function. My concern is the novelty of this manuscript. 

For the introduction, more information about the vascular reactivity index is needed. Why was the vascular reactivity index (VRI) chosen as a method to measure endothelial dysfunction? What is the advantage of this method and why didn't you also measure, for example, circulating endothelin and thrombomodulin as an indicator of endothelial function? 

Author Response

Comments and Suggestions for Authors

The manuscript describes a clinical study in which the association between serum indoxyl sulfate levels and endothelial function in non-dialysis CKD patients was investigated. Already multiple studies (in vitro and in vivo studies) have shown that indoxyl sulfate induces deleterious effects on the endothelial function. My concern is the novelty of this manuscript. 

For the introduction, more information about the vascular reactivity index is needed. Why was the vascular reactivity index (VRI) chosen as a method to measure endothelial dysfunction? What is the advantage of this method and why didn't you also measure, for example, circulating endothelin and thrombomodulin as an indicator of endothelial function? 

Ans: Thanks for your recommendations. We used VRI as a method to indicate endothelial function because it gave a good opportunity for direct, real-time assessment of microvascular function in a non-invasive manner instead of measuring surrogate serum levels of biomarkers, such as endothelin, C-reactive protein, homocysteine, plasminogen activator inhibitor-1, etc. Moreover, measuring VRI is an easy and relatively cost-effective technique which offers the opportunity of future clinical implications to find early endothelial damage. We revised the description of in Introduction section about why we chose the method of digital thermal monitoring as “Recently, Naghavi et al. conducted a study in a registry of 6084 patients comparing different methods of measuring functional endothelial dysfunction, including digital thermal monitoring, flow-mediated dilatation, peripheral arterial tonometry and photoplethysmography. Digital thermal monitoring was shown to provide equivalent ability of measurements compared to other methods and was easier of use and applicability to medical staff or patients.

Round 2

Reviewer 1 Report

The authors have adequately addressed the queries.

Reviewer 4 Report

The authors have adequately answered to my concern.